# Tick Populations and Molecular Analysis of *Anaplasma* Species in Ticks from the Republic of Korea

**DOI:** 10.3390/microorganisms11040820

**Published:** 2023-03-23

**Authors:** Min-Goo Seo, Haeseung Lee, Badriah Alkathiri, KyuSung Ahn, Seung-Hun Lee, SungShik Shin, Seulgi Bae, Kyoo-Tae Kim, Min Jang, Sang-Kwon Lee, Yun Sang Cho, Kyung-Yeon Eo, Oh-Deog Kwon, Dongmi Kwak

**Affiliations:** 1College of Veterinary Medicine, Kyungpook National University, Daegu 41566, Republic of Korea; 2Animal and Plant Quarantine Agency, Gimcheon 39660, Republic of Korea; 3College of Veterinary Medicine, Chungbuk National University, Cheongju 28644, Republic of Korea; 4Department of Parasitology, College of Veterinary Medicine, Chonnam National University, Gwangju 61186, Republic of Korea; 5Department of Animal Health and Welfare, College of Healthcare and Biotechnology, Semyung University, Jecheon 27136, Republic of Korea

**Keywords:** *Anaplasma phagocytophilum*, *Anaplasma phagocytophilum*-like *Anaplasma* spp., *Anaplasma bovis*, *Anaplasma capra*, questing tick

## Abstract

The present study was performed to survey the dominant tick populations and molecularly determine the pathogenic agents of anaplasmosis in ticks from Gyeongsang, Republic of Korea. A total of 3825 questing ticks were collected by the flagging method from 12 sites near animal farms in Gyeongsang from March to October 2021. A molecular genomic study was performed with ticks stored in 70% ethanol to detect *Anaplasma* genes by the previously described method. The monthly incidence of ticks varied by developmental stages, i.e., nymphs, adults, and larvae, and each of their populations peaked in May, March, and October, respectively. The predominant tick species were *Haemaphysalis longicornis, Haemaphysalis* sp., *Haemaphysalis flava, Ixodes nipponensis*, and *Amblyomma testudinarium* in order. To determine the *Anaplasma* infection rate, collected ticks were pooled into 395 groups. The minimum infection rate (MIR) of *Anaplasma* was 0.7% (27 pools). That of *A. phagocytophilum* was highest (23 pools, MIR 0.6%), followed by *A. phagocytophilum*-like *Anaplasma* spp. clade B (2 pools, MIR 0.1%), *A. bovis* (1 pool, MIR 0.1%), and *A. capra* (1 pool, MIR 0.1%), respectively. In this study, five species of ticks, including unidentified *Haemaphysalis* species, were collected in 12 survey sites in Gyeongsang, but their prevalence was somewhat different according to the tick species and survey sites. Further, the incidence rate (6.8%) of 4 *Anaplasma* spp. was not as high in tick pools. However, the results of this study may offer a basis for future epidemiological research and risk assessment of tick-borne diseases.

## 1. Introduction

Numerous emerging tick-borne diseases were present in ticks, humans, and animals before they were recognized as the causal agents of clinical illnesses [1]. Numerous factors, such as tick incidence, human activity, duration of tick attachment, climatic and geographic factors, and the biological stage, affect the ability of ticks to attach to humans and transmit pathogens [2]. Owing to the gradual changes in temperature in the Republic of Korea from a temperate to a subtropical region, the Korean Peninsula may soon experience increases in tick populations. Ticks depend on environmental factors to survive, and climate change provides conducive conditions [3].

Diseases associated with *Anaplasma* species have affected the health and production of animals for more than a century and are currently posing a serious threat to livestock [4]. Ticks transmit *Anaplasma* in nature, and there are numerous vertebrate hosts and infection sources for ticks, humans, and animals [4]. The sporozoan parasites of the genus *Anaplasma* have been taxonomically known as several verified species, i.e., *A. phagocytophilum, A. marginale, A. centrale, A. bovis, A. platys, A. ovis*, and *Candidatus* Anaplasma species. These species have a somewhat different host cell tropism. The predilection site of *A. phagocytophilum* is neutrophils [5], *A. marginale, A. centrale*, and *A. ovis* prefer erythrocytes [6], *A. bovis* and *A. platys* specifically parasitize in monocytes [7] and platelets [8], respectively. A recently recognized *Anaplasma* species, *Anaplasma capra*, may infect endothelial cells [9]. Additionally, it is crucial to distinguish between pathogenic *A. phagocytophilum* and closely related *A. phagocytophilum*-like *Anaplasma* spp. (APL), which are currently thought to be nonpathogenic and do not cause clinical symptoms in infected animals [10].

Numerous *Anaplasma* spp. in tick populations have been detected in the Republic of Korea, such as *A. capra* and *A. bovis* from ticks parasitizing water deer (*Hydropotes inermis argyropus*) [11]; *A. capra, A. bovis*, APL clade A, and APL clade B from ticks parasitizing cattle [12]; *A. bovis* from ticks parasitizing native Korean goats [13]; and *A. phagocytophilum* and *A. platys* from small wild-caught mammals or by dragging/flagging [14]. In other countries, several *Anaplasma* spp. have also been identified in tick populations, such as *A. marginale, A. platys*, and *A. capra* from ticks parasitizing cattle and goats in China [15]; *A. phagocytophilum* by flagging in Austria [16]; and *A. bovis* by flagging in Canada [17].

To predict the origin of infections and offer risk assessments for tick-borne diseases, the present study was performed to study the prevalence of tick populations and the incidence rates and risk factors for pathogenic *Anaplasma* species in ticks, which were collected from 12 sites near animal farms in Gyeongsang (six in Gyeongnam and six in Gyeongbuk), Republic of Korea.

## 2. Materials and Methods

### 2.1. Ethical Approval

Approval from Kyungpook National University’s Institutional Animal Care and Use Committee was not required for the present study, which was conducted in 2021. The collected samples from questing ticks in the environment in this study did not cause hazard to any animals. The collected ticks did not include endangered species. Specific approval for each collection site was not needed because the sites were not located within national parks or protected regions.

### 2.2. Tick Collection

Questing ticks were collected by the flagging method, in which the vegetation is swept with a flannel cloth [18], from 12 sites near animal farms in Gyeongsang from March to October 2021. The survey sites were administratively in Gyeongbuk (GB), Gyeongnam (GN), and Ulsan Metropolitan City (UMC), Republic of Korea. The six sites in GB were Bonghwa (BH), Chilgok (CG), Gumi (GM), Uiseong (US,), Yeongcheon (YC), and Yeongdeok (YD). The four in GN were Geochang (GC), two in Goseong (GS), and Haman (HA), and the remaining two were UMC regions (Table 1 and Figure 1). Unfed ticks were collected once per month. In the two provinces, the GS and ULS regions had two collection sites each, and each of the other eight regions was used as a collection site, totaling 12 tick collection sites close to livestock farms (cattle, deer, horse, and goat farms). The collected ticks were stored in 70% ethanol.

Tick species and developmental stages were identified based on morphological characteristics using a microscope according to taxonomic key [19]. Each identified tick was pooled according to species, developmental stage, survey period, and collection site. Adults and nymphs were classified at the species level, whereas larvae were identified to the genus level because of morphological similarities. The number of ticks in the pools was one to 10 nymphs and one to 50 larvae, and the adults were individually examined.

### 2.3. DNA Extraction and PCR Detection

Genomic DNA was extracted using a DNeasy Blood & Tissue Kit (Qiagen, Hilden, Germany) according to the manufacturer’s instructions. PCR was performed using an AccuPower HotStart PCR Premix Kit (Bioneer, Daejeon, Republic of Korea). Nested PCR was performed to detect the genus *Anaplasma* by amplifying the 16S rRNA gene, as previously described [20] using the primer pairs EE1/EE2 and EE3/EE4, which produced an amplicon of 924–926 bp. A sample of *A. phagocytophilum* detected in cattle in the Republic of Korea [21] was included as a positive control, and a sample without a DNA template was used as a negative control.

### 2.4. Sequencing and Phylogenetic Analyses

All the PCR-positive products with EE3/EE4 primers were sent to Macrogen (Daejeon, Republic of Korea) for Sanger sequencing. The sequences obtained in the present study and previously reported in GenBank were aligned and analyzed using the multiple sequence alignment program CLUSTAL Omega (v. 1.2.1, Bioweb, Ferndale, WA, USA). Among all the long-aligned nucleotide sequences, unnecessary sequences in the front and back were deleted based on the sequences detected in this study using BioEdit (v. 7.2.5, Bioedit, Manchester, UK). Sites containing gaps or having ambiguous alignment were also removed prior to phylogenetic analysis. Phylogenetic analysis was performed using the maximum likelihood method with the Kimura two-parameter distance model in molecular evolutionary genetics analysis (v. 7.0, Mega software solutions, Madhurawadha, India). The aligned sequences of *Anaplasma* 16S rRNA were pairwise compared to determine homology. The stability of the obtained trees was estimated using bootstrap analysis with 1000 replicates.

### 2.5. Statistical Analysis

To evaluate the correlation between *Anaplasma* prevalence and five different tick species, data were statistically analyzed using the chi-square test using the GraphPad Prism analytical software package (v. 5.04, GraphPad Software, Inc., La Jolla, CA, USA). Statistical significance was set at *p* < 0.05.

## 3. Results

### 3.1. Prevalence of Ticks by the Verified Species

A total of 3825 questing ticks (1798 from GB and 2027 from GN and UMC) belonging to five species, i.e., *Haemaphysalis* sp., *Haemaphysalis longicornis, Haemaphysalis flava, Ixodes nipponensis*, and *Amblyoma testudinarium*, were collected from 12 survey sites (Table 2). All larvae belonging to the genus *Haemaphysalis* were regarded as *Haemaphysalis* sp. (Yamaguti) because the *H. longicornis* and *H. flava* larvae are morphologically indistinguishable. *Haemaphysalis* spp. and *I. nipponensis* were identified in both provinces, whereas *A. testudinarium* was only identified in the GS region of GN province.

*H. longicornis* was the most abundant species (*n* = 2405; GB 1103, GN 1302), followed by *Haemaphysalis* sp. (*n* = 1176; GB 582, GN 594), *H. flava* (*n* = 217; GB 96, GN 121), *I. nipponensis* (*n* = 22; GB 17, GN 5), and *A. testudinarium* (*n* = 5; GB 0, GN 5).

We collected 86 adult ticks (2.3%), including 34 females (0.9%) and 52 males (1.4%), 2563 nymphs (67.0%), and 1176 larvae (30.8%).

### 3.2. Incidence of Anaplasma Genes from Tick Pools

Among 3825 collected ticks, 395 pools were tested using PCR, and *Anaplasma*-positive ticks were detected in 27 pools (6.8%). The minimum infection rate (MIR) was 0.7 (Table 3).

*I. nipponensis* had the highest prevalence of 30.0% (6/20 pools, MIR: 27.3), followed by *Haemaphysalis* sp. (25.8%, 8/31 pools, MIR: 0.7) and *H. longicornis* (4.4%, 13/294 pools, MIR: 0.5). No *Anaplasma*-positive cases were reported in *H. flava* or *A. testudinarium*. The *Anaplasma* prevalence was significantly different in the different tick species (chi-square test, χ^2^ = 40.73, df = 4, *p* < 0.0001).

The GB and GN provinces showed 5.7% (11/192 pools, MIR: 0.6) and 7.9% (16/203 pools, MIR: 0.8) positive pools, respectively. Among them, the ULS (13.5%, 5/37 pools, MIR: 1.1), GM (10.8%, 4/37 pools, MIR: 1.4), CG (10.5%, 4/38 pools, MIR: 1.1), and GC (9.5%, 2/21 pools, MIR: 0.7) regions had a higher prevalence than that of other regions.

The prevalence of nymphs, adult females and males, and larva stages were 5.2% (13/278 pools, MIR: 0.5), 8.8% (3/34 pools, MIR: 8.8), 5.8% (3/52 pools, MIR: 5.8), and 25.8% (8/31 pools, MIR: 0.7), respectively.

The number of adult females and males, nymphs, and larvae was highest in June (32.4%, 11/34), March (40.4%, 21/52), May (29.4%, 753/2563), and October (36.6%, 430/1176), respectively (Figure 2a). Temporally, the highest peak in tick incidence was observed in May (20.2%, 771/3825), when nymphs were highly prevalent; a second peak was observed in August (15.3%, 584/3825) when larvae were beginning to be prevalent.

The largest number of positive tick pools was observed in July (25.9%, 7/27, MIR: 0.18) and March (18.5%, 5/27, MIR: 0.13) (Figure 2b).

### 3.3. Molecular and Phylogenetic Analyses

Phylogenetic analysis revealed that *Anaplasma* spp. identified in the present study were *A. phagocytophilum*, APL clade B, *A. bovis*, and *A. capra* (Table 4 and Figure 3). Of the 27 positive pools, the *A. phagocytophilum* incidence was the highest (23 pools, 85.2%), followed by APL clade B (2 pools, 7.5%), *A. bovis* (1 pool, 3.7%), and *A. capra* (1 pool, 3.7%).

The two representative sequences of *A. phagocytophilum* 16S rRNA sequences in the present study shared 99.3% identity with each other. They were also 98.3–100% identical to the 16S rRNA sequences of previously reported *A. phagocytophilum* isolates. We determined that the two sequences of the APL clade B 16S rRNA sequences shared 99.7% identity with each other and 98.8–100% identity with the 16S rRNA sequences of previously reported APL clade B isolates. One *A. capra* sequence in the present study shared 99.5–99.8% identity with the 16S rRNA sequences of previously reported *A. capra* isolates. Similarly, one *A. bovis* 16S rRNA sequence shared 99.5–99.8% identity with other *A. bovis* isolates. Among the sequences obtained in the present study, representative sequences used in the phylogenetic analysis were submitted to the GenBank database (accession numbers: OP535541–OP535545).

## 4. Discussion

In this study, more than five species, including unidentified *Haemaphysalis* sp., were collected from 12 survey sites in Gyeongsang, Republic of Korea. Among them, *H. longicornis* (62.9%) was most prevalent, and *Haemaphysalis* sp. (30.8%), *H. flava* (5.7%), *I. nipponensis* (0.6%), and *A. testudinarium* (0.1%) followed in descending order. These findings were more or less similar with those of Seo et al. [22]. They reported the tick prevalence among *H. longicornis* (56.7%), *Haemaphysalis* sp. (40.5%), *H. flava* (1.7%), *I. nipponensis* (0.1%), and *A. testudinarium* (0.3%) from a nationwide survey in the Republic of Korea, respectively.

Questing ticks near animal farms were surveyed monthly between March and October 2021. We evaluated the geographical distribution of tick species and found that most ticks were collected in the HA (13.2%), GS (2 sites with an average of 10.1%), and CG (9.5%) regions. Compared with other regions, the HA and GS regions are located at low latitudes in the Republic of Korea with higher temperatures and more precipitation, which are ideal conditions for tick survival [23]; thus, these regions had a higher distribution of tick populations than that in other regions. *Haemaphysalis* spp. were widely distributed in the surveyed regions, and *I. nipponensis* was distributed in five regions (four in GB and one in GN). However, *A. testudinarium* was only distributed in the GS region, located at the lowest latitude in the surveyed regions. Therefore, ecological and environmental factors may impact the regional variations in tick distribution.

Ticks spend most of their lives in the natural environment, except when feeding; hence, environmental factors impact their biology. Changes in the environment and season, particularly humidity and temperature, affect tick development, survival, distribution, and, subsequently, the tick-borne disease risk [24]. According to a previous study, peaks in adult, nymph, and larval tick populations were generally observed from June to August, May to June, and August to September in the Republic of Korea [18,22], consistent with the findings of the present study, in which the highest peaks in nymphs, adults, and larval populations were observed in May, March, and October, respectively.

We performed molecular detection and phylogenetic analysis of *Anaplasma* (6.8%) in ticks. Normally, *Anaplasma* spp. are more prevalent in adult ticks than those in nymphs and larvae because adults have a higher probability of coming into contact with an *Anaplasma*-infected host. In the present study, nymphs (48.2%) were more prevalent in *Anaplasma* infections than adults (22.2%) and larvae (29.6%). Temporally, the number of positive tick pools in adults was highest in July. The number of positive tick pools in nymphs was highest in July. Infected larval ticks were detected between August and October. The MIR of *Anaplasma* in ticks was relatively high in March (0.13) and July (0.18), probably due to the high risk of *Anaplasma* infection in those months. Therefore, caution is needed when engaging in outdoor activities around this period.

In the present study, *H. longicornis* (13 pools, three female adults, and 10 nymphs) was the most prevalent tick species host to *Anaplasma*, followed by *I. nipponensis* (six pools, three male adults, and three nymphs), and *Haemaphysalis* spp. (eight pools and eight larvae). The prevalence of *Anaplasma* was significantly different among the tick species (*p* < 0.0001). However, *H. flava* and *A. testudinarium* were not the main vector for *Anaplasma* spp. As the detection of *Anaplasma* in ticks does not indicate the capacity of the ticks to act as competent vectors, further studies are needed to establish whether these species are *Anaplasma* vectors. Tick-borne diseases are transmitted through transstadial or transovarial routes. In the present study, 31 pools (1176 larvae) were tested for *Anaplasma* spp., and eight pools (MIR: 0.7) were positive. Thus, the larvae generally transmitted *Anaplasma* transovarially (25.8%). Further studies are needed to determine whether adult ticks can transovarially transmit *the Anaplasma* species.

Molecular analyses in previous studies have detected several *Anaplasma* spp. in various hosts in the Republic of Korea, including *A. capra* (2.9%, 5/173 pools) and *A. bovis* (2.3%, 4/173 pools) from ticks parasitizing water deer (*Hydropotes inermis argyropus*) [11]; *A. capra* (4.7%, 27/576), *A. bovis* (2.3%, 13/576), APL clade A (1.4%, 11/576), and APL clade B (0.5%, 3/576) from ticks parasitizing cattle [12]; *A. bovis* (2.5%, 1/40 pools) from ticks parasitizing native Korean goats [13]; *A. capra* (0.4%, 5/1219) and *A. bovis* (1.0%, 12/1219) from cattle [21]; *A. phagocytophilum* (2.1%, 16/764) and APL clade A (2.6%, 20/764) from cattle [25]; *A. phagocytophilum* (2.6%, 13/510) from dogs [26]; *A. phagocytophilum* (0.2%, 1/627) from horses [27]; *A. phagocytophilum* (1.8%, 31/1696) and *A. bovis* (1.7%, 29/1696) from horses [28]; and *A. phagocytophilum* in patients and biting ticks [29]. Analyses of the nucleotide sequences from a variety of genes have been used to report the genetic diversity of the genus *Anaplasma* [4]. For example, 16S rRNA gene analysis are well supported by comparable *groESL* clades, as well as biological and antigenic features. The *groESL* sequences provide support for the divisions shown by the 16S rRNA gene sequences and provide evidence of polymorphisms that may be accidental or may show subtleties of evolutionary selection. Therefore, the current analysis of the 16S rRNA gene may be supported by additional sequence analyses of conserved and semi-conserved genes (such as *gltA*), whole genome analysis, and analysis of new strains [30]. Because only the 16S rRNA gene was used in this study, there are limitations to the molecular identification of *Anaplasma* species. In the present study, we detected four *Anaplasma* species from questing ticks by amplifying the 16S rRNA gene, including *A. phagocytophilum* (0.6% MIR), APL clade B (0.1% MIR), *A. capra* (0.1% MIR), and *A. bovis* (0.1% MIR). Among them, *A. phagocytophilum* (85.2%) was the most prevalent; seven *Haemaphysalis* spp. larvae, one *H. longicornis* adult, nine *H. longicornis* nymphs, three *I. nipponensis* adults, and three *I. nipponensis* nymphs were detected. *A. phagocytophilum* is a serious zoonotic pathogen in humans and animals [29]. *A. capra* was detected in one adult *H. longicornis*. In 2018, *A. capra* was detected for the first time in Korean cattle in the GN province [21]. *A. capra* is a potential zoonotic pathogen from ticks parasitizing animals in the Republic of Korea. However, it remains unclear whether *A. capra* is pathogenic to humans and animals; therefore, additional research is needed to clarify the pathogenicity of this emerging *Anaplasma* species. *A. bovis* (one *H. longicornis* adult) and APL clade B (one *H. longicornis* nymph and one *Haemaphysalis* spp. larva) were also detected but were not considered zoonotic pathogens. Further studies are needed to determine the pathogenicity of the *Anaplasma* species.

Our study reveals the geographical and temporal distribution of several tick species and their *Anaplasma* infections. In this study, five species of ticks including unidentified *Haemaphysalis* species were collected in 12 survey sites of Gyeongsang, but their prevalence was somewhat different according to the tick species and survey sites. As mentioned above, four different *Anaplasma* species were previously detected in several animals, humans, and animal blood-feeding ticks. To the best of our knowledge, this study is the first to investigate the presence of *A. phagocytophilum*, APL clade B, *A. bovis*, and *A. capra* in questing unfed ticks in an environment near animal farms in the Republic of Korea. Further, the incidence rate (6.8%) of four *Anaplasma* spp. was not as high in tick pools. However, the results of this study may offer a basis for future epidemiological research and risk assessment of tick-borne diseases. Further studies in larger regions with livestock, ticks, and wild animals are also needed to understand biology of tick-pathogen infections and to prevent the transmission of tick-borne diseases and establish effective control strategies.

## Figures and Tables

**Figure 1 microorganisms-11-00820-f001:**
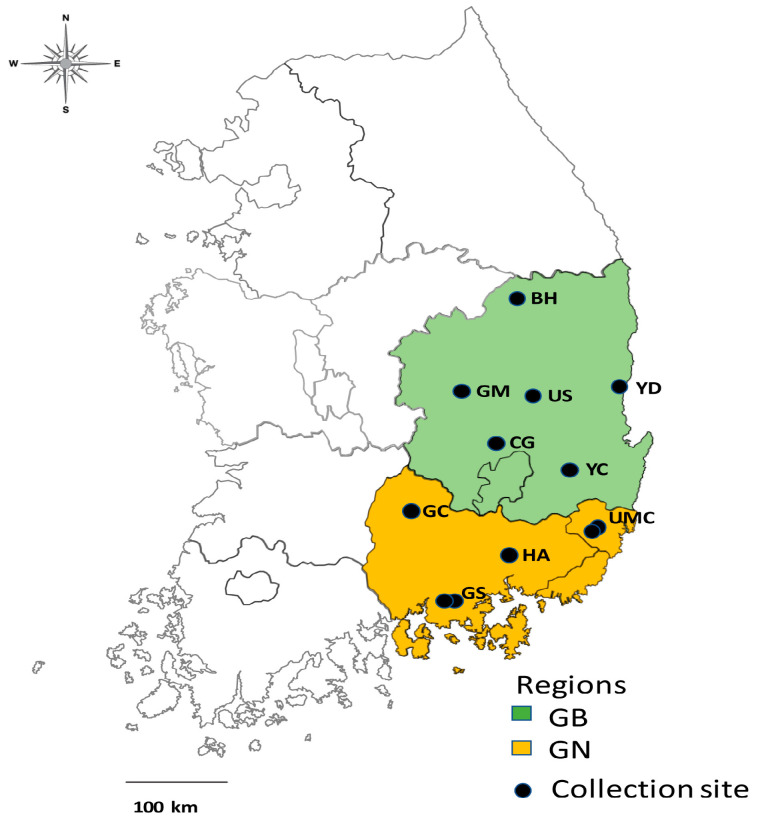
Map of 12 survey sites in Gyeongsang, Republic of Korea, showing six [Bonghwa (BH), Chilgok (CG), Gumi (GM), Uiseong (US), Yeongcheon (YC), and Yeongdeok (YD)] in Gyeongbuk (GB), four [Geochang (GC), two in Goseong (GS) and Haman (HA)] in Gyeongnam (GN), and two in Ulsan Metropolitan City (UMC).

**Figure 2 microorganisms-11-00820-f002:**
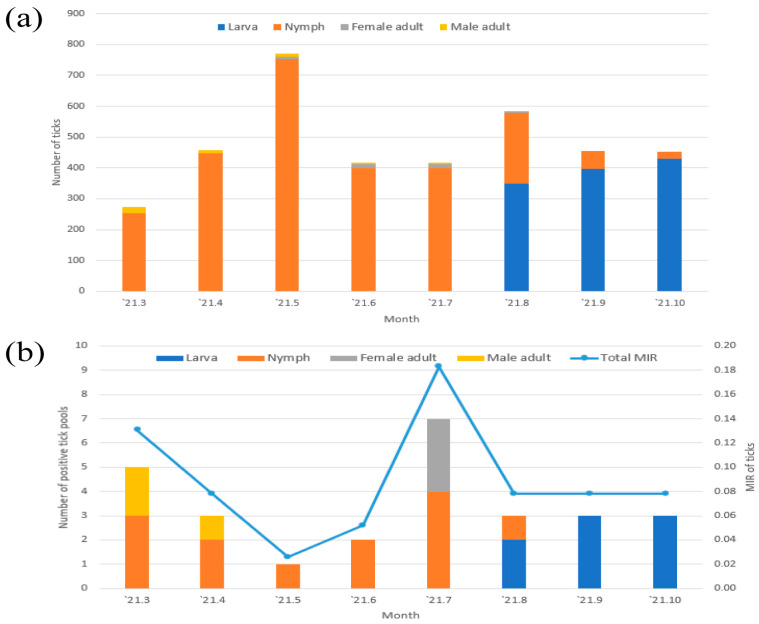
(**a**) Temporal distribution of the total (nymphs, adults [males and females], and larvae) population of ticks. (**b**) The number of positive tick pools and minimum infection rate (MIR) of *Anaplasma*-infected ticks collected in the Republic of Korea. MIR = (number of positive pools of ticks/total number of ticks tested × 100).

**Figure 3 microorganisms-11-00820-f003:**
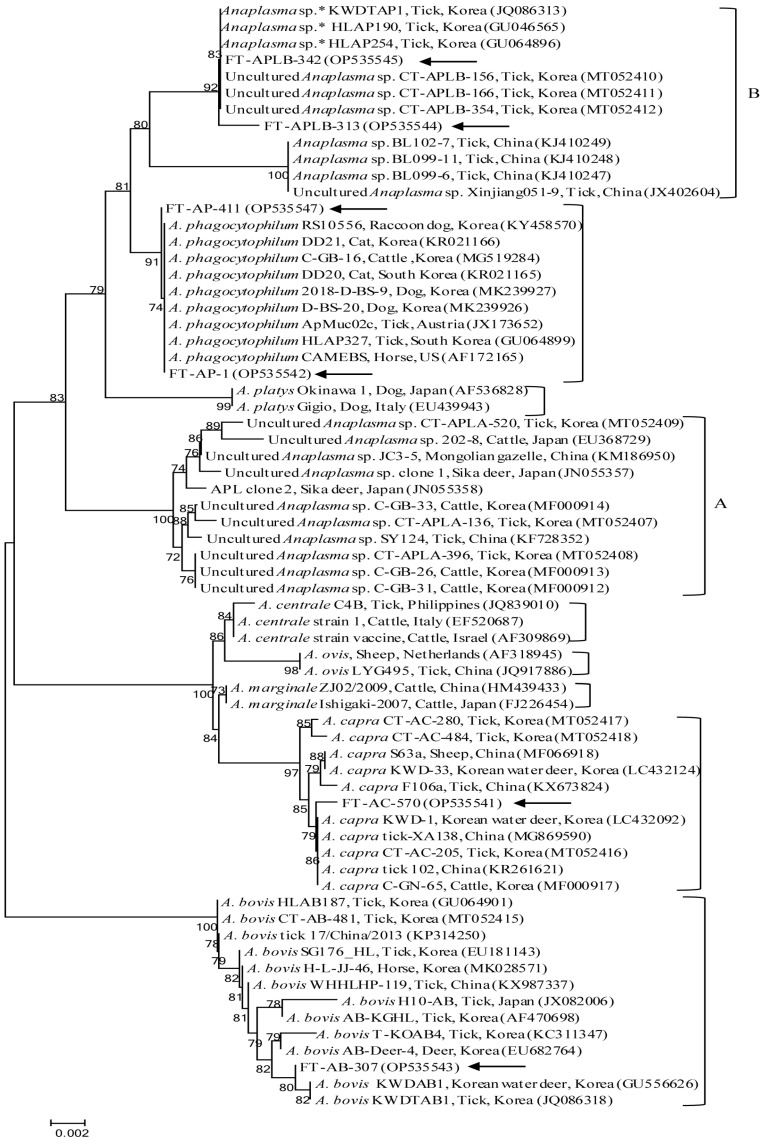
Phylogenetic tree of *Anaplasma* spp. based on sequences of the 16S rRNA gene. The tree was constructed using the maximum likelihood method with the Kimura two-parameter distance model. Black arrows show the sequences analyzed in the present study. GenBank accession numbers of other sequences presented with the sequence name. *Anaplasma* sp. * deposited in the GenBank as *A. phagocytophilum*. Numbers on the branches correspond to mean bootstrap support at 1000 replicates. The scale and the scale bar show phylogenetic distance. A and B, two clades of *Anaplasma phagocytophilum*-like *Anaplasma* spp.

**Table 1 microorganisms-11-00820-t001:** Tick collection sites in the Republic of Korea.

Region		GPS Coordinates
Gyeongbuk (GB)	Bonghwa (BH)	36°53′35.55″ N 128°38′44.47″ E
	Chilgok (CG)	36°00′58.62″ N 128°29′09.54″ E
	Gumi (GM)	36°19′41.85″ N 128°16′32.10″ E
	Uiseong (US)	36°25′57.05″ N 128°44′44.82″ E
	Yeongcheon (YC)	35°50′44.46″ N 128°59′02.38″ E
	Yeongdeok (YD)	36°20′16.97″ N 129°21′57.81″ E
Gyeongnam (GN)	Geochang (GC)	35°47′35.23″ N 127°57′50.43″ E
	Goseong (GS)	35°00′04.13″ N 128°16′17.48″ E
	Goseong (GS)	35°00′06.78″ N 128°15′17.83″ E
	Haman (HA)	35°20′13.90″ N 128°34′06.50″ E
Ulsan Metropolitan City (UMC)		35°30′51.46″ N 129°10′03.78″ E
		35°32′18.93″ N 129°10′18.64″ E

**Table 2 microorganisms-11-00820-t002:** Distribution of tick species according to the region and developmental stage in the Republic of Korea.

Species	Stage	No. of Ticks Collected by Region ^1^										Total
		GB						GN			UMC	
		BH	CG	GM	US	YC	YD	GC	GS	HA		
*Haemaphysalis* sp.	Larva	100	112	120	150	60	40	150	154	90	200	1176
*Haemaphysalis longicornis*	Nymph	201	234	181	170	145	150	130	510	371	258	2350
	Adult (M) ^2^	3	2	1	3	1	1	1	7	3	1	23
	Adult (F) ^2^	2	1	2	3	1	2	3	7	7	4	32
	Subtotal	206	237	184	176	147	153	134	524	381	263	2405
*Haemaphysalis flava*	Nymph	44	8	7	8	8	8	0	89	24	2	198
	Adult (M)	1	1	4	0	5	1	0	1	5	0	18
	Adult (F)	1	0	0	0	0	0	0	0	0	0	1
	Subtotal	46	9	11	8	13	9	0	90	29	2	217
*Ixodes nipponensis*	Larva	0	0	0	0	0	0	0	0	0	0	0
	Nymph	0	2	3	3	1	0	0	0	2	0	11
	Adult (M)	0	3	3	0	2	0	0	0	3	0	11
	Adult (F)	0	0	0	0	0	0	0	0	0	0	0
	Subtotal	0	5	6	3	3	0	0	0	5	0	22
*Amblyomma testudinarium*	Larva	0	0	0	0	0	0	0	0	0	0	0
	Nymph	0	0	0	0	0	0	0	4	0	0	4
	Adult (M)	0	0	0	0	0	0	0	0	0	0	0
	Adult (F)	0	0	0	0	0	0	0	1	0	0	1
	Subtotal	0	0	0	0	0	0	0	5	0	0	5
Total		352	363	321	337	223	202	284	773	505	465	3825

^1^ Among 10 regions, the six survey sites in Gyeongbuk (GB) were Bonghwa (BH), Chilgok (CG), Gumi (GM), Uiseong (US), Yeongcheon (YC), and Yeongdeok (YD); the four survey sites in Gyeongnam (GN) were Geochang (GC), two in Goseong (GS), and Haman (HA); and the two survey sites were Ulsan Metropolitan City (UMC). ^2^ M, male; F, female.

**Table 3 microorganisms-11-00820-t003:** *Anaplasma* prevalence in ticks based on tick species, developmental stage, and region.

Species	Stage	Tested Tick (Pool) ^2^	No. Positive Tick Pool/Tick Pool Tested in a Region ^1^										Total	MIR ^3^	*p*-Value ^4^
			GB						GN			UMC			
			BH	CG	GM	US	YC	YD	GC	GS	HA				
*Haemaphysalis* sp.	Larva	1176 (31)	0/3	1/3	1/3	0/3	0/2	0/2	2/4	2/4	0/2	2/5	8/31	0.7	<0.0001
*Haemaphysalis longicornis*	Nymph	2350 (239)	0/21	2/24	0/19	2/17	0/15	1/15	0/13	1/51	1/38	3/26	10/239	0.4	
	Adult (M) ^5^	23 (23)	0/3	0/2	0/1	0/3	0/1	0/1	0/1	0/7	0/3	0/1	0/23	0	
	Adult (F) ^5^	32 (32)	0/2	0/1	1/2	0/3	0/1	0/2	0/3	1/7	1/7	0/4	3/32	9.4	
	Subtotal	2405 (294)	0/26	2/27	1/22	2/23	0/17	1/18	0/17	2/65	2/48	3/31	13/294	0.5	
*Haemaphysalis flava*	Nymph	198 (28)	0/5	0/2	0/3	0/1	0/2	0/1	0	0/10	0/3	0/1	0/28	0	
	Adult (M)	18 (18)	0/1	0/1	0/4	0	0/5	0/1	0	0/1	0/5	0	0/18	0	
	Adult (F)	1 (1)	0/1	0	0	0	0	0	0	0	0	0	0/1	0	
	Subtotal	217 (47)	0/7	0/3	0/7	0/1	0/7	0/2	0	0/11	0/8	0/1	0/47	0	
*Ixodes nipponensis*	Larva	0	0	0	0	0	0	0	0	0	0	0	0		
	Nymph	11 (9)	0	0/2	2/2	0/3	0/1	0	0	0	1/1	0	3/9	27.3	
	Adult (M)	11 (11)	0	1/3	0/3	0	0/2	0	0	0	2/3	0	3/11	27.3	
	Adult (F)	0	0	0	0	0	0	0	0	0	0	0	0		
	Subtotal	22 (20)	0	1/5	2/5	0/3	0/3	0	0	0	3/4	0	6/20	27.3	
*Amblyomma testudinarium*	Larva	0	0	0	0	0	0	0	0	0	0	0	0		
	Nymph	4 (2)	0	0	0	0	0	0	0	0/2	0	0	0/2	0	
	Adult (M)	0	0	0	0	0	0	0	0	0	0	0	0		
	Adult (F)	1 (1)	0	0	0	0	0	0	0	0/1	0	0	0/1	0	
	Subtotal	5 (3)	0	0	0	0	0	0	0	0/3	0	0	0/3	0	
Total		3825 (395)	0/36	4/38	4/37	2/30	0/29	1/22	2/21	4/83	5/62	5/37	27/395	0.7	

^1^ Among 10 regions, six survey sites were in Gyeongbuk (GB) were Bonghwa (BH), Chilgok (CG), Gumi (GM), Uiseong (US), Yeongcheon (YC), and Yeongdeok (YD); four survey sites were in Gyeongnam (GN) were Geochang (GC), two in Goseong (GS), and Haman (HA); and two survey sites were in Ulsan Metropolitan City (UMC). ^2^ Adult ticks were not pooled. ^3^ MIR, minimum infection rate (number of positive pools of ticks total number of ticks tested × 100). ^4^ Significant (*p* < 0.05) correlation with infection. ^5^ M, male; F, female.

**Table 4 microorganisms-11-00820-t004:** The minimum infection rate of *Anaplasma* species in the Republic of Korea.

Species	No. of Positive Tick Pool/MIR ^2^			
	Larva	Nymph	Adult	Total
	(*n* = 1176)	(*n* = 2563)	(*n* = 86)	(*n* = 3825)
*Anaplasma phagocytophilum*	7/0.6	12/0.5	4/4.7	23/0.6
APL ^1^ clade B	1/0.1	1/0.1	0	2/0.1
*Anaplasma capra*	0	0	1/1.2	1/0.1
*Anaplasma bovis*	0	0	1/1.2	1/0.1
Total	8/0.7	13/0.5	6/7.0	27/0.7

^1^ APL, *Anaplasma phagocytophilum*-like *Anaplasma* spp. ^2^ MIR, minimum infection rate (number of positive pools of ticks total number of ticks tested × 100).

## Data Availability

Data supporting the conclusions of this article are included within the article. The newly generated sequences were submitted to the GenBank database under the accession numbers OP535541–OP535545. The datasets used and/or analyzed during the present study are available from the corresponding author upon reasonable request.

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
