# Peer review of "Tick Populations and Molecular Analysis of Anaplasma Species in Ticks from the Republic of Korea"

_microorganisms, 2023, doi:10.3390/microorganisms11040820_

Round 1

Reviewer 1 Report

Sites and coordinates of where ticks were collected should be presented in a table format for a neat presentation.

Sentence 108-109: Each identified tick was pooled according to species.

Explain the pooling method in detail.

Sentence 122: All the PCR-positive products were sent to Macrogen (Daejeon, the Republic of 122 Korea), for nucleotide sequencing.

Were these PCR amplicons sent to Macrogen? Were the product sequenced on both ends, specify that this was a sanger sequencing?

Well written, however molecular identification of Anaplasma species based solely on 16S rRNA may have limitations and this should be highlighted in the study

Author Response

Response to Reviewer 1

Sites and coordinates of where ticks were collected should be presented in a table format for a neat presentation.

Response: We appreciate your relevant comment. According to your suggestions, we changed the sentence as “Six sites in GB were Bonghwa (BH), Chilgok (CG), Gumi (GM), Uiseong (US,), Yeongcheon (YC), and Yeongdeok (YD). Four in GN were Geochang (GC), 2 in Goseong (GS) and Haman (HA), and remain 2 were UMC regions (Table 1 and Figure 1).” at lines 91-94. Table 1 was added for coordinates.

Sentence 108-109: Each identified tick was pooled according to species.

Explain the pooling method in detail.

Response: We appreciate your relevant comment. Tick samples were pooled according to species, developmental stage, survey period, and collection site (1-50 larvae per pool, 1-30 nymphs per pool, and 1 adult individually used). Because of morphological similarities, adults and nymphs were classified to the species level, whereas larvae were identified to the genus level. For instance, all Haemaphysalis sp. was larva stage and total number is 1,176. Number of ticks and pools are described in below table.  According to your suggestions, we changed the sentence more clearly as “Each identified tick was pooled according to species, developmental stage, survey period, and collection site. Adults and nymphs were classified to the species level, whereas larvae were identified to the genus level because of morphological similarities. The number of ticks in the pools was 1 to 10 nymphs and 1 to 50 larvae, and the adults were individually examined.” at lines 106-110.

Species

Stage

No. of ticks collected by region 1

Total

GB

GN

UMC

BH

CG

GM

US

YC

YD

GC

GS

HA

Haemaphysalis sp.

Individual Larva

100

112

120

150

60

40

150

154

90

200

1,176

Pooling group

50,30,20

50,12,50

50,30,40

50,50,50

20,40

20,20

50,50,30,20

50,24,30,50

50,40

50,30,50,30,40

1,176

Number of pools

3

3

3

3

2

2

4

4

2

5

31

Sentence 122: All the PCR-positive products were sent to Macrogen (Daejeon, the Republic of Korea), for nucleotide sequencing.

Were these PCR amplicons sent to Macrogen? Were the product sequenced on both ends, specify that this was a sanger sequencing?

Response: We appreciate your relevant comment. We detected total 27 positive pooled samples by PCR. All 27 PCR products were sent to Macrogen for nucleotide sequencing using EE3 and EE4 primers. Macrogen did purification of PCR products and Sanger sequencing. According to your suggestions, we changed the sentence more clearly as “All the PCR-positive products with EE3/EE4 primers were sent to Macrogen (Daejeon, Republic of Korea), for sanger sequencing.” at lines 121-122.

Well written, however molecular identification of Anaplasma species based solely on 16S rRNA may have limitations and this should be highlighted in the study

Response: We appreciate your critical comment. In this study, molecular identification of Anaplasma species based solely on 16S rRNA may have limitations. In the near future, we will perform molecular analysis of other genes such as heat shock protein (groEL), complete citrate synthase (gltA), and major surface protein 4 (msp4). And we will also practice new advanced techniques so that we can apply them. According to your suggestions, we added the sentence as “Analyses of the nucleotide sequences from a variety of genes have been used to report on the genetic diversity of the genus Anaplasma [30]. For example, 16S rRNA gene analysis are well supported by comparable groESL clades, as well as biological and antigenic features. The groESL sequences provide support for the divisions shown by the 16S rRNA gene sequences and provide evidence of polymorphisms that may be accidental or may show subtleties of evolutionary selection. Therefore, the current analysis of the 16S rRNA gene may be supported by additional sequence analyses of conserved and semi-conserved genes (such as gltA), whole genome analysis, and analysis of new strains [31]. Because only the 16S rRNA gene was used in this study, there are limitations to the molecular identification of Anaplasma species.” at lines 284-294.

Reviewer 2 Report

The manuscript “Tick Populations and Molecular Analysis of Anaplasma Species in Ticks from Republic of Korea” deserves to be published in the Journal Microorganisms after addressing the following concerns:

Line 87-88 Please, describes the methodology of the flagging method or at least reference it.

Lines 158-159: Do the pools have the number of larvae/nymphs collected detailed in Table 2? In M&M, it is mentioned that the pools contained up to 10 nymphs and 50 larvae (lines 109-110), and e.g. in the Haemaphysalis pools, there are 1176 larvae /50 larvae per pool= 24 pools, and in Table 2, 31 pools are mentioned; With various data on larvae and nymphs, the number of pools does not correspond, or what was said in lines 109-110, were the pools for identification and a different number was used to perform the PCR? Confused!

Line 270: What does that percentage mean? The number of Anaplasma-positive larvae, the percentage of larvae transmitting Anaplasma previously reported in the literature, ¿?...

Author Response

Response to Reviewer 2

Comments and Suggestions for Authors

The manuscript “Tick Populations and Molecular Analysis of Anaplasma Species in Ticks from Republic of Korea” deserves to be published in the Journal Microorganisms after addressing the following concerns:

Line 87-88 Please, describes the methodology of the flagging method or at least reference it.

Response: We appreciate your relevant comment. According to your suggestion, we changed sentence as “Questing ticks were collected by the flagging method, in which the vegetation is swept with a flannel cloth [18], from 12 sites near animal farms in Gyeongsang, from March to October 2021.” at lines 88-90.

Lines 158-159: Do the pools have the number of larvae/nymphs collected detailed in Table 2? In M&M, it is mentioned that the pools contained up to 10 nymphs and 50 larvae (lines 109-110), and e.g. in the Haemaphysalis pools, there are 1176 larvae /50 larvae per pool= 24 pools, and in Table 2, 31 pools are mentioned; With various data on larvae and nymphs, the number of pools does not correspond, or what was said in lines 109-110, were the pools for identification and a different number was used to perform the PCR? Confused!

Response: We appreciate your relevant comment. Tick samples were pooled according to species, developmental stage, survey period, and collection site (1-50 larvae per pool, 1-30 nymphs per pool, and 1 adult individually used). Adults and nymphs were classified to the species level, whereas larvae were identified to the genus level because of morphological similarities. For instance, all Haemaphysalis sp. were larva stage and total number is 1,176. Number of ticks and pools are described in below table. According to your suggestions, we changed the sentence more clearly as “Each identified tick was pooled according to species, developmental stage, survey period, and collection site. Adults and nymphs were classified to the species level, whereas larvae were identified to the genus level because of morphological similarities. The number of ticks in the pools was 1 to 10 nymphs and 1 to 50 larvae, and the adults were individually examined.” at lines 106-110.

Species

Stage

No. of ticks collected by region 1

Total

GB

GN

UMC

BH

CG

GM

US

YC

YD

GC

GS

HA

Haemaphysalis sp.

Individual Larva

100

112

120

150

60

40

150

154

90

200

1,176

Pooling group

50,30,20

50,12,50

50,30,40

50,50,50

20,40

20,20

50,50,30,20

50,24,30,50

50,40

50,30,50,30,40

1,176

Number of pools

3

3

3

3

2

2

4

4

2

5

31

Line 270: What does that percentage mean? The number of Anaplasma-positive larvae, the percentage of larvae transmitting Anaplasma previously reported in the literature, ¿?...

Response: We appreciate your relevant comment. Those percentage is infection rate of pathogens in various hosts such as animals and ticks (larva, nymph, and adult stage). According to your suggestion, we added sentence as “Molecular analyses in previous studies detected several Anaplasma spp. in various hosts in the Republic of Korea, including A. capra (2.9%, 5/173 pools) and A. bovis (2.3%, 4/173 pools) from ticks parasitizing water deer (Hydropotes inermis argyropus) [11]; A. capra (4.7%, 27/576), A. bovis (2.3%, 13/576), APL clade A (1.4%, 11/576), and APL clade B (0.5%, 3/576) from ticks parasitizing cattle [12]; A. bovis (2.5%, 1/40 pools) from ticks parasitizing native Korean goats [13]; A. capra (0.4%, 5/1219) and A. bovis (1.0%, 12/1219) from cattle [21]; A. phagocytophilum (2.1%, 16/764) and APL clade A (2.6%, 20/764) from cattle [25]; A. phagocytophilum (2.6%, 13/510) from dogs [26]; A. phagocytophilum (0.2%, 1/627) from horses [27]; A. phagocytophilum (1.8%, 31/1696) and A. bovis (1.7%, 29/1696) from horses [28]; A. phagocytophilum in patient and biting tick [29].” at lines 275-284.

Reviewer 3 Report

After reading the paper I have following comments and suggestion for authors:

Please remove „(‘do’ means Province)”

L29 Revise according to alphabetic order

Use abbreviated name of pathogens after first mention of the genus in the text

L55 the authors do not list all pathogenic Anaplasma species, please include also Anaplasma Candidatus species in this sentence too.

L66 water deer, please add Latin name of this animal

L74 “to know” I suggest to revise to “to study”

L87 this information “A total of 3,825 questing” is not necessary here. I suggest to revise this sentence and include total number of collected ticks only in results section.

L97 “Fig. 1” revise according to journal requirements “Figure 1”

L108 “identification standards” revise to “taxonomic key”

L124 the authors state “and the alignment was corrected using BioEdit”. Could you describe in more details this correction? what was the correction?

L126 why do you deciced to use ML method to construct phylogenetic trees? Did you construct trees using other methods too, such as MP, NJ? had they a similar topology?

L126 What model was used to construct the trees? This is most important here.

L126 I also suggest the authors to use in the future analysis newer version of MEGA instead of v 7.0. MEGA 11 is free available now.

L128 “The aligned sequences were analyzed using a similarity matrix” Not clear for me. Could you explain please?

L132 Please explain criteria of statistical test selection? Why chi-square?

Also, in my opinion the authors could perform more advanced analysis of molecular investigation based on results obtained in MEGA. You can calculates p-distances between Anaplasma seq obtained in this study and those available in GenBank (included in the tree). You can also calculate statistical differences in the number of nucleotide change position. This will make your results more solid.

Figure 3. I can see some clades drawn by the authors but only some of them are named. Why?   

My main comment on Figure 3 is that the authors should include much more sequences to construct the tree (especially when we based on 16S RNA and want to know taxonomical position of studied seq). What was the criteria of seq selection? I my opinion there should be ca min. 10 other seq per one seq obtained in this study and in the tree should be included all Anaplasma species available in GenBank. Finally this tree should be ca 50-6o seq.  

Rickettsia hulinii was used as out group to root the tree, right? I strongly suggest to reconstruct the tree.

Also, as you performared molecular analysis based on 16S, please refer to this paper:

Dumler, J. Stephen, et al. "Reorganization of genera in the families Rickettsiaceae and Anaplasmataceae in the order Rickettsiales: unification of some species of Ehrlichia with Anaplasma, Cowdria with Ehrlichia and Ehrlichia with Neorickettsia, descriptions of six new species combinations and designation of Ehrlichia equi and'HGE agent'as subjective synonyms of Ehrlichia phagocytophila." International journal of systematic and evolutionary microbiology 51.6 (2001): 2145-2165.

Author Response

Response to Reviewer 3

Comments and Suggestions for Authors

After reading the paper I have following comments and suggestion for authors:

Please remove „(‘do’ means Province)”

Response: Yes, we did as suggested in manuscript.

L29 Revise according to alphabetic order

Response: We appreciate your relevant comment. But this sentence means that Haemaphysalis longicornis was most prevalent and Haemaphysalis sp., Haemaphysalis flava, Ixodes nipponensis, and Amblyomma testudinarium in order. So, we do not need to revise according to alphabetic order.

Use abbreviated name of pathogens after first mention of the genus in the text

Response: Yes, we did as suggested in manuscript at line 31.

L55 the authors do not list all pathogenic Anaplasma species, please include also Anaplasma Candidatus species in this sentence too.

Response: We appreciate your relevant comment. According to your suggestion, we added sentence as “The sporozoan parasites of the genus Anaplasma have been taxonomically known as several verified species, i.e., A. phagocytophilum, A. marginale, A. centrale, A. bovis, A. platys, A. ovis and Candidatus Anaplasma species.” at lines 55-58.

L66 water deer, please add Latin name of this animal

Response: Yes, we did as suggested in manuscript at lines 67-68.

L74 “to know” I suggest to revise to “to study”

Response: Yes, we did as suggested in manuscript at line 75.

L87 this information “A total of 3,825 questing” is not necessary here. I suggest to revise this sentence and include total number of collected ticks only in results section.

Response: Yes, we did as suggested in manuscript at line 88.

L97 “Fig. 1” revise according to journal requirements “Figure 1”

Response: Yes, we did as suggested in manuscript at line 94.

L108 “identification standards” revise to “taxonomic key”

Response: Yes, we did as suggested in manuscript at line 106.

L124 the authors state “and the alignment was corrected using BioEdit”. Could you describe in more details this correction? what was the correction?

Response: We appreciate your relevant comment. First, in order to compare the sequence detected in this study with other sequences in Genbank, we were aligned and analyzed using the multiple sequence alignment program CLUSTAL Omega. And then among all the long-aligned nucleotide sequences, unnecessary sequences in the front and back were deleted based on the sequences detected in this study using BioEdit. Sites containing gaps or having ambiguous alignment were also removed prior to phylogenetic analysis. According to your suggestion, we changed the sentence as “The sequences obtained in the present study and previously reported in GenBank were aligned and analyzed using the multiple sequence alignment program CLUSTAL Omega (v. 1.2.1, Bioweb, Ferndale, WA, USA). Among all the long-aligned nucleotide sequences, unnecessary sequences in the front and back were deleted based on the sequences detected in this study using BioEdit (v. 7.2.5, Bioedit, Manchester, UK). Sites containing gaps or having ambiguous alignment were also removed prior to phylogenetic analysis.” at lines 122-128.

L126 why do you deciced to use ML method to construct phylogenetic trees? Did you construct trees using other methods too, such as MP, NJ? had they a similar topology?

Response: We appreciate your relevant comment. The maximum likelihood (ML) method has several advantages over the neighbor-joining (NJ) and maximum parsimony (MP) methods in the phylogenetic analysis of pathogen genes. ML takes into account the evolutionary model of sequence evolution, which is particularly important for pathogen genes that undergo rapid evolution and may be subject to selective pressures, recombination events, or other complex evolutionary scenarios. In contrast, NJ and MP do not consider the evolutionary model and may produce less accurate results. ML is statistically consistent, meaning it converges to the true tree as the amount of data increases. This is important for pathogen genes where the amount of available data may be limited, and accuracy is crucial for identifying the source and transmission routes of the pathogen. NJ is not statistically consistent, while MP can be statistically consistent under certain conditions. ML can handle missing data and can provide estimates of the uncertainty of the tree topology, such as bootstrap values. This is important for pathogen genes where the quality of the sequence data may vary, and missing data may be common. NJ and MP may not be able to handle missing data or may produce less reliable estimates of uncertainty. ML allows testing of alternative hypotheses of evolution through statistical tests, such as the likelihood ratio test. This is important for pathogen genes where different evolutionary scenarios may need to be evaluated to identify the most likely source and transmission routes. NJ and MP do not provide a formal way of testing alternative hypotheses. Overall, the ML method provides a more accurate and rigorous approach to phylogenetic analysis of pathogen genes than NJ and MP, particularly in terms of modeling sequence evolution, statistical consistency, robustness, and hypothesis testing. However, MP may still be a useful method for some pathogen genes, particularly if the sequence data is limited, and the evolutionary scenarios are relatively simple. Therefore, phylogenetic analysis was performed using the ML method in this study.

L126 What model was used to construct the trees? This is most important here.

Response: We appreciate your relevant comment. We used the maximum likelihood method with the Kimura two-parameter distance model. According to your suggestions, we added the sentence as “Phylogenetic analysis was performed using the maximum likelihood method with the Kimura two-parameter distance model in molecular evolutionary genetics analysis (v. 7.0, Mega software solutions, Madhurawadha, India).” at lines 128-131.

L126 I also suggest the authors to use in the future analysis newer version of MEGA instead of v 7.0. MEGA 11 is free available now.

Response: We appreciate your relevant comment. According to your suggestion, we will use a new version of MEGA in the future.

L128 “The aligned sequences were analyzed using a similarity matrix” Not clear for me. Could you explain please?

Response: We appreciate your relevant comment. We aligned sequences of Anaplasma 16S rRNA were pairwise compared to determine homology. Comparative analysis of the 16S rRNA nucleotide sequences from the obtained samples with the other Anaplasma isolates included in the GenBank database were analyzed with a similarity matrix. The upper matrix shows percent identity between the partial sequences of the Anaplasma 16S rRNA gene. The lower matrix presents the number of differences in nucleotide bases. According to your suggestion, we changed the sentence more clearly as “The aligned sequences of Anaplasma 16S rRNA were pairwise compared to determine homology.” at lines 131-132.

L132 Please explain criteria of statistical test selection? Why chi-square?

Response: We appreciate your relevant comment. We want to know statistical difference of Anaplasma prevalence and 5 different tick species (Haemaphysalis sp., Haemaphysalis longicornis, Haemaphysalis flava, Ixodes nipponensis, and Amblyoma testudinarium). Normally, chi-squared test is a statistical hypothesis test used in the analysis of contingency tables when the sample sizes are large. In simpler terms, this test is primarily used to examine whether two categorical variables (two dimensions of the contingency table) are independent in influencing the test statistic (values within the table). So, we used chi-square test in this study. According to your suggestion, we added the sentence as “To evaluate the correlation between Anaplasma prevalence and 5 different tick species, data were statistically analyzed using the chi-square test using GraphPad Prism analytical software package (v. 5.04, GraphPad Software, Inc., La Jolla, CA, USA).” at lines 135-137.

Also, in my opinion the authors could perform more advanced analysis of molecular investigation based on results obtained in MEGA. You can calculates p-distances between Anaplasma seq obtained in this study and those available in GenBank (included in the tree). You can also calculate statistical differences in the number of nucleotide change position. This will make your results more solid.

Response: We appreciate your critical comment. But, in this case, we could not perform those advanced analyses. In the near future, we will practice new advanced techniques so that you can apply them in other research.

Figure 3. I can see some clades drawn by the authors but only some of them are named. Why?  

Response: We appreciate your relevant comment. We detected several Anaplasma such as A. phagocytophilum, APL clade B, A. bovis and A. capra in this study. So, we denote each group by drawing parentheses for ease of viewing each species. In particular, the APL clades are labeled as A and B for easy identification.

My main comment on Figure 3 is that the authors should include much more sequences to construct the tree (especially when we based on 16S RNA and want to know taxonomical position of studied seq). What was the criteria of seq selection? I my opinion there should be ca min. 10 other seq per one seq obtained in this study and in the tree should be included all Anaplasma species available in GenBank. Finally this tree should be ca 50-6o seq. 

Response: We appreciate your relevant comment. The criteria of sequence selection are related isolate of hosts (such as tick, animal and human) and regions (such as Korea, China and Japan). According to your suggestion, we added other related Anaplasma species sequences in GenBank and reconstructed the tree in Figure 3.

Rickettsia hulinii was used as out group to root the tree, right? I strongly suggest to reconstruct the tree.

Response: We appreciate your relevant comment. According to your suggestion, we deleted out group and reconstructed the tree including related Anaplasma spp. in Figure 3.

Also, as you performared molecular analysis based on 16S, please refer to this paper:

Dumler, J. Stephen, et al. "Reorganization of genera in the families Rickettsiaceae and Anaplasmataceae in the order Rickettsiales: unification of some species of Ehrlichia with Anaplasma, Cowdria with Ehrlichia and Ehrlichia with Neorickettsia, descriptions of six new species combinations and designation of Ehrlichia equi and'HGE agent'as subjective synonyms of Ehrlichia phagocytophila." International journal of systematic and evolutionary microbiology 51.6 (2001): 2145-2165.

Response: We appreciate your critical comment. In this study, molecular identification of Anaplasma species based solely on 16S rRNA may have limitations. In the near future, we will perform molecular analysis of other genes such as heat shock protein (groEL), complete citrate synthase (gltA), and major surface protein 4 (msp4). And we will also practice new advanced techniques so that we can apply them. According to your suggestions, we added the sentence as “Analyses of the nucleotide sequences from a variety of genes have been used to report on the genetic diversity of the genus Anaplasma [30]. For example, 16S rRNA gene analysis are well supported by comparable groESL clades, as well as biological and antigenic features. The groESL sequences provide support for the divisions shown by the 16S rRNA gene sequences and provide evidence of polymorphisms that may be accidental or may show subtleties of evolutionary selection. Therefore, the current analysis of the 16S rRNA gene may be supported by additional sequence analyses of conserved and semi-conserved genes (such as gltA), whole genome analysis, and analysis of new strains [31]. Because only the 16S rRNA gene was used in this study, there are limitations to the molecular identification of Anaplasma species.” at lines 284-294.

Round 2

Reviewer 3 Report

The authors revised paper according to my suggestions or provided detailed replied replies. In my opinion the manuscript may move through the publishing process